# Downregulation of miR-17-92 Cluster by PERK Fine-Tunes Unfolded Protein Response Mediated Apoptosis

**DOI:** 10.3390/life11010030

**Published:** 2021-01-06

**Authors:** Danielle E. Read, Ananya Gupta, Karen Cawley, Laura Fontana, Patrizia Agostinis, Afshin Samali, Sanjeev Gupta

**Affiliations:** 1Discipline of Pathology, Cancer Progression and Treatment Research Group, Lambe Institute for Translational Research, School of Medicine, National University of Ireland-Galway, H91 TK33 Galway, Ireland; read.danielle@gmail.com; 2Discipline of Physiology, School of Medicine, National University of Ireland-Galway, H91 TK33 Galway, Ireland; ananya.gupta@nuigalway.ie; 3Apoptosis Research Centre, School of Natural Sciences, National University of Ireland Galway, H91 TK33 Galway, Ireland; k.cawley1@gmail.com (K.C.); afshin.samali@nuigalway.ie (A.S.); 4Ragon Institute of MGH, MIT and Harvard, Cambridge, 02138 MA, USA; lfontana1@mgh.harvard.edu; 5Cell Death Research and Therapy Group, Department of Cellular and Molecular Medicine, KU Leuven, 3000 Leuven, Belgium; patrizia.agostinis@kuleuven.be; 6VIB Center for Cancer Biology Research, 3000 Leuven, Belgium

**Keywords:** unfolded protein response, ER stress, apoptosis, microRNAs, miR-17-92, ATF4, NRF2, CHOP

## Abstract

An important event in the unfolded protein response (UPR) is activation of the endoplasmic reticulum (ER) kinase PERK. The PERK signalling branch initially mediates a prosurvival response, which progresses to a proapoptotic response upon prolonged ER stress. However, the molecular mechanisms of PERK-mediated cell death are not well understood. Here we show that expression of the primary miR-17-92 transcript and mature miRNAs belonging to the miR-17-92 cluster are decreased during UPR. We found that miR-17-92 promoter reporter activity was reduced during UPR in a PERK-dependent manner. Furthermore, we show that activity of the miR-17-92 promoter is repressed by ectopic expression of ATF4 and NRF2. Promoter deletion analysis mapped the region responding to UPR-mediated repression to a site in the proximal region of the miR-17-92 promoter. Hypericin-mediated photo-oxidative ER damage reduced the expression of miRNAs belonging to the miR-17-92 cluster in wild-type but not in PERK-deficient cells. Importantly, ER stress-induced apoptosis was inhibited upon miR-17-92 overexpression in SH-SY5Y and H9c2 cells. Our results reveal a novel function for ATF4 and NRF2, where repression of the miR-17-92 cluster plays an important role in ER stress-mediated apoptosis. Mechanistic details are provided for the potentiation of cell death via sustained PERK signalling mediated repression of the miR-17-92 cluster.

## 1. Introduction

Physiological and pathological conditions that interfere with homeostasis of the endoplasmic reticulum (ER) can lead to protein misfolding. The accumulation of misfolded proteins within the ER is known as ER stress. The unfolded protein response (UPR) constitutes a signal transduction pathway that responds to the accumulation of misfolded proteins in the ER. The UPR orchestrates an increase in ER-folding capacity through transcriptional induction of ER folding, lipid biosynthesis and ER associated degradation (ERAD) machinery along with a concomitant decrease in the rate of protein synthesis through selective mRNA degradation and translational repression [1]. The UPR is therefore cytoprotective, allowing cells to adapt to perturbations that impinge on ER protein folding. However, during severe and prolonged ER stress, the UPR can become cytotoxic and induce apoptosis [2,3].

In metazoans, ER stress is detected by three ER resident proteins: activating transcription factor 6α (ATF6), protein kinase RNA-like ER kinase (PERK) and inositol-requiring enzyme 1α (IRE1) [4]. During resting conditions, the activity of PERK, IRE1 and ATF6 are kept in check by the binding of immunoglobulin protein (BiP)/glucose-regulated protein 78 kDa (GRP78) to their luminal domain. The accumulation of unfolded proteins sequesters BiP/GRP78 away from PERK, IRE1 and ATF6 leading to their activation. PERK-mediated phosphorylation of eukaryotic translation initiation factor 2α (eIF2α) on the alpha subunit at Ser51 leads to translational attenuation [5]. Whilst phosphorylation of eIF2α inhibits general translation initiation, it paradoxically increases translation of activating transcription factor 4 (ATF4), which induces the transcription of genes involved in restoration of ER homeostasis [5]. The activity of nuclear factor (erythroid-derived 2)-like 2 (NRF2), a b-ZIP cap “n” collar transcription factor, can also be induced by activated PERK. NRF2 is normally bound to its negative regulator Kelch-like ECH-associated protein (KEAP1) in the cytoplasm [6]. Activated PERK phosphorylates NRF2 leading to its dissociation from KEAP1 [7] and translocation to the nucleus where it binds to the antioxidant response element (ARE) within the promotors of its target genes. The endoribonuclease activity of IRE1 is responsible for the nonconventional splicing of transcription factor X-box binding protein 1 (XBP1), which controls the transcription of chaperones and genes involved in ERAD [8]. In addition to XBP1 splicing, active IRE1 can also degrade a set of RNAs, including mRNAs, rRNAs and miRNAs, via a mechanism known as regulated IRE1-dependent decay (RIDD) [9]. In response to ER stress, ATF6 traffics to the Golgi complex where S1P and S2P proteases cleave the cytosolic and transmembrane domains. The processed form of ATF6 (N-terminal transcriptional regulatory domain) translocates to the nucleus thereby regulating the transcription of genes involved in ER homeostasis, such as ER chaperones and ERAD components [1].

Although we understand the activation and general function of UPR components the molecular mechanisms of ER stress-induced cell death are not well understood. microRNAs (miRNAs) are a family of short (20–23 nucleotide), endogenous, single-stranded RNA molecules that regulate gene expression in a sequence-specific manner [10]. miRNAs are generated from primary RNA transcripts and are processed by the microprocessor complex within the nucleus to generate small hairpin RNA (precursor miRNA), which are subsequently exported to the cytoplasm. In the cytoplasm precursor-miRNA molecules undergo Dicer-mediated processing, thus generating mature miRNAs [11]. The mature miRNA assembles into the ribonucleoprotein silencing complexes (RISCs) and guides the silencing complex to specific mRNA molecules [12]. The main function of miRNAs is to direct posttranscriptional regulation of gene expression, typically by binding to the 3’ untranslated region (UTR) of cognate mRNAs and inhibiting their translation and/or stability [13,14]. miRNAs have been implicated in nearly all developmental and pathological processes in animals, including tissue morphogenesis, cell proliferation, apoptosis and major signalling pathways [12,13]. miRNAs play an important role in the regulation of key cell survival and cell death decisions [15]. During the past few years, work from several groups has revealed that all three branches of the UPR regulate specific subsets of miRNAs [15,16]. The modes of regulation include induction/repression by UPR-regulated transcription factors such as ATF6, XBP1, ATF4, CCAAT-enhancer-binding protein homologous protein (CHOP) and NRF2 as well as IRE1-mediated miRNA degradation [4,17]. The outcome of UPR-dependent miRNA expression is fine tuning of the ER homeostasis to modulate the adaptation to stress and regulation of cell fate [15,17,18]. Global approaches in several cellular contexts have revealed that UPR regulates the expression of many miRNAs that play an important role in the regulation of life and death decisions during UPR [16,19].

The miR-17-92 cluster comprises a group of six miRNAs on chromosome 13 that is transcribed as a single polycistronic unit [20]. Amplification and overexpression of the miR-17-92 cluster has been documented in B-cell lymphomas, lung cancer and gastric cancer [20]. Although several studies have characterised the oncogenic activities of miR-17-92, loss of heterozygosity at the locus that harbours human miR-17-92 has been reported [21,22]. For example, deletion of the miR-17-92 cluster has been reported in 16.5% of ovarian cancers, 21.9% of breast cancers and 20.0% of melanomas [22]. Indeed, miR-17-92 exhibits context-dependent tumour suppressive effects in ER-positive and oncogenic effects in triple negative breast cancer [23]. The transcription factors c-MYC and E2F3 induce the expression of the miR-17-92 cluster [24] whereas TP53 and AML1 down-regulate its expression [25,26]. Enforced expression of miR-17-92 cluster miRNAs in an Eμ-myc transgenic mouse model of B cell lymphoma accelerates disease onset and progression [27]. Deletion of the miR-17-92 cluster results in smaller embryos and immediate postnatal death of all animals, and is associated with severe lung hypoplasia and a defective ventricular septum [28]. However, in contrast to the wealth of information about the biological effects of the miR-17-92 cluster, little is known about its regulation.

In this study we describe the role of the miR-17-92 cluster in ER stress responses. Our data indicate that expression of the miR-17-92 cluster is reduced during conditions of ER stress in a variety of cell types. Ectopic expression of ATF4 or NRF2 leads to reduced expression of miRNAs belonging to the miR-17-92 cluster. We further show that the miR-17-92 cluster is repressed by the PERK-dependent transcription factors ATF4 and NRF2. We provide evidence that repression of the miR-17-92 cluster contributes to ER stress-mediated apoptosis. Taken together our results suggest a role for downregulation of the miR-17-92 cluster in fine tuning cell fate during the ER stress response.

## 2. Materials and Methods

### 2.1. Cell Culture and Treatments

MEFs, H9c2, MDA-MB231, 293T, PC12, SH-SY5Y and MCF-7 cells were maintained in Dulbecco’s modified medium (DMEM) supplemented with 10% foetal calf serum (FCS), 100 U/mL penicillin and 100 mg/mL streptomycin at 37 °C with 5% CO_2_. PC12 cells were cultured in DMEM supplemented with 10% heat inactivated horse serum, 5% foetal bovine serum and 1% penicillin/streptomycin at 37 °C with 5% CO_2_. Cells were treated with thapsigargin and tunicamycin for the times indicated. Glucose deprivation of H9c2 cells was achieved by replacing the serum- and glucose-containing DMEM with serum- and glucose free-DMEM containing 2-deoxyglucose (1 mM) for 24 h. All reagents were from Sigma (Sigma-Aldrich, Arklow, Ireland) unless otherwise stated.

### 2.2. Plasmid Constructs

The miR-17-92 cluster promoter reporter constructs containing −829 to +2922 bp (pGL4-17-92FL) and +1492 to +2922 bp (pGL4-17-92/1) were supplied by Dr. Laura Fontana, Ragon Institute of MGH, MIT and Harvard, Cambridge MA, USA [26,29] while the construct containing +2626 to +2922 bp (pGL4-17-92) was from Dr. Scott Hammond, University of North Carolina, NC, USA [30]. The expression plasmids for ATF4, wild type PERK and K618A PERK were obtained from Dr. David Ron, University of Cambridge, Cambridge, UK (Addgene plasmid #21814 and #21815). The expression vector for wild type NRF2 was from Dr. Alan Diehl, University of Pennsylvania, USA [7]. The expression vector for wild-type C/EBP homologous protein (CHOP) was from Dr. Andreas Strasser, WEHI, Australia [31]. For the lentiviral expression of miR-17-92, a 1 kb fragment spanning the miR-17-92 cluster was amplified using pCXN2-miR17-92 plasmid [32] as a template and cloned in to pCDH-CMV-EF1-RFP. Transient transfections were carried out using Lipofectamine 2000 (Bio-Sciences, Co Dublin Ireland) according to the manufacturer’s protocol.

### 2.3. Generation of Stable Cell Lines

We generated stable subclones of H9c2 and SH-SY5Y expressing the miR-17-92 cluster by transducing cells with pCDH-CMV-miR-17-92-EF1-RFP or the corresponding control pCDH-CMV-EF1-RFP. Cells were transduced with the lentivirus using polybrene (5 μg/mL) to increase the transduction efficiency. The subclones expressing miR-17-92 cluster were obtained by sorting on the basis of red fluorescent protein (RFP) using a fluorescence-activated cell sorting (FACS) AriaII cell sorter to attain >90% RFP positivity in the selected population.

### 2.4. RNA Extraction, RT-PCR and Real Time RT-PCR

Total RNA was isolated using RNeasy kit (Qiagen, Manchester, UK) according to the manufacturer’s instructions. Reverse transcription (RT) was carried out with 2 μg RNA and Oligo dT (Invitrogen) using 20 U Superscript II Reverse Transcriptase (Bio-Sciences, Co Dublin Ireland). The real-time PCR method, used to determine the induction of UPR target genes, has been described previously [33,34]. Briefly, cDNA products were mixed with 2× TaqMan master mixes and 20× TaqMan Gene Expression Assays (Bio-Sciences, Co Dublin Ireland) and subjected to 40 cycles of PCR using a StepOnePlus instrument (Bio-Sciences, Co Dublin Ireland). Relative expression was evaluated using the 2^–∆∆Ct^ method.

### 2.5. Measurement of miRNA Levels Using TaqMan qRT-PCR Assays

Total RNA was reverse transcribed using the TaqMan miRNA Reverse Transcription Kit and miRNA-specific stem-loop primers (Bio-Sciences, Co Dublin, Ireland) in a small-scale RT reaction (comprised of 0.19 μL of H_2_O, 1.5 μL of 10× Reverse-Transcription Buffer, 0.15 μL of 100 mM dNTPs, 1.0 μL of Multiscribe Reverse-Transcriptase (50 Units/μL) and 5.0 μL of input RNA (20 ng/μL)); components other than the input RNA were prepared as a larger volume master mix, using a Tetrad2 Peltier Thermal Cycler (BioRad) at 16 °C for 30 min, 42 °C for 30 min and 85 °C for 5 min. For primary miR-17-92 (Cat#Hs03295901_pri), miR-17 (Cat#002308), miR-18a (Cat#002422), miR-19a (Cat#000395), miR-19b (Cat#000396), miR-20a (Cat#000580), miR-92a (Cat#000431), snoRNA (Cat#001718) and U6 snRNA (Cat#001973), 4.0 μL of RT product was combined with 16.0 μL of PCR assay reagents (comprised of 5.0 μL of H_2_O, 10.0 μL of TaqMan Universal PCR Master Mix (2×), no AmpErase UNG and 1.0 μL of TaqMan miRNA Assay) to generate a PCR of 20.0 μL in total volume. Real-time PCR was carried out using an Applied BioSystems 7900HT thermocycler at 95 °C for 10 min, followed by 40 cycles of 95 °C for 15 s and 60 °C for 1 min. Data were analysed with SDS Relative Quantification Software version 2.2.2 (Bio-Sciences, Co Dublin Ireland), with the automatic Ct setting for assigning baseline and threshold for Ct determination.

### 2.6. Luciferase Reporter Assays

In promoter assays, MCF-7 cells were transfected with 0.8 μg of firefly luciferase vectors (empty pGL4 or pGL4-miR-17-92 promoter reporters), in combination with a *Renilla* luciferase vector (0.2 μg) as the internal control. At 24 h post-transfection, cells were treated with thapsigargin or tunicamycin for 24 h. Firefly luciferase and *Renilla* luciferase activities were measured 48 h after transfection at 560 nm using a 10 s luminescence protocol with a Wallac plate reader and then normalised for *Renilla* luciferase activity.

### 2.7. Apoptotic Nuclei Measurements

Cells were grown on coverslips and after treatment, cells were fixed in (10% *v/v*) formaldehyde for 10 min at room temperature. After washing with phosphate buffered saline (PBS) cells were mounted on glass slides in a mountant with DAPI (Vectashield_Cat#H-1200). Nuclei were visualised using an Olympus BX61 fluorescence microscope. Apoptotic nuclei (condensed, fragmented, intensely stained) were counted and presented as a percentage of the total nuclei. At least 100 cells were counted per well, and all treatments were performed in triplicate.

### 2.8. Western Blotting

Cells were washed once in ice-cold PBS and lysed in whole cell lysis buffer (20 mM HEPES pH 7.5, 350 mM NaCl, 0.5 mM EDTA, 1 mM MgCl_2_, 0.1 mM EGTA and 1% NP-40) after the stipulated time of treatments and boiled at 95 °C with Laemmli’s sodium dodecyl sulphate–polyacrylamide gel electrophoresis (SDS-PAGE) sample buffer for 5 min. Protein concentration was determined by the Bradford method. Equal amounts (20 μg/lane) of protein samples were run on an SDS polyacrylamide gel. The proteins were transferred onto nitrocellulose membrane and blocked with 5% milk in PBS-0.05% Tween. The membrane was incubated with the primary antibody for cleaved caspase-3 (ISIS, Cat#9664) or β-Actin (Sigma, Cat#A-5060) for 2 h at room temperature or overnight at 4 °C. The membrane was washed three times with PBS-0.05% Tween and further incubated in the appropriate horseradish peroxidase-conjugated secondary antibody (Pierce) for 90 min. Signals were detected using Western Lightening Plus ECL (SGR Scientific Limited, Dublin, Ireland).

### 2.9. Statistical Analysis

The data were expressed as the mean ± SD for three independent experiments. Differences between the treatment groups were assessed using Two-tailed paired student’s *t*-tests. As a parametric procedure the paired *t*-test makes several assumptions. The key assumptions for paired *t*-test are: (i) the dependent variable should be approximately normally distributed and (ii) the dependent variable should not contain any outliers. Since we had a very small number of extreme data points we assumed that the distributions of all data sets analysed are normal. Given that we found that standard deviation was substantially lower than the mean, it is a reasonable assumption. The values with a *p* < 0.05 were considered statistically significant.

## 3. Results

### 3.1. Expression of the pri-miR-17-92 is Down-Regulated upon Induction of ER Stress

In preliminary experiments, the relative abundance of miRNAs comprising the Sanger miRBase database (Release 11.0) were analysed by microarray (LC sciences, Houston, TX, USA) using RNA from H9c2 cells during conditions of ER stress (Appendix A, in the Appendix A). We observed downregulation of all six miRNAs belonging to the miR-17-92 cluster in H9c2 cells treated with thapsigargin (TG) or tunicamycin (TM). TG (an inhibitor of the sacroplasmic/endoplasmic reticulum Ca^2+^-ATPase (SERCA) pump) and TM (an inhibitor of N-linked glycosylation) both lead to accumulation of misfolded proteins in the ER and initiate the UPR [35]. Since the six miRNAs belonging to the miR-17-92 cluster are derived from the primary transcript of the miR-17-92 gene, we reasoned that the miR-17-92 gene is transcriptionally regulated during ER stress. Next, we evaluated expression of the primary miR-17-92 transcript during conditions of ER stress. We observed that upon treatment of H9c2 cells with TG the level of GRP78 increased in a time-dependent manner. However, under similar conditions, levels of the primary miR-17-92 transcript were reduced (Figure 1A). Glucose deprivation is one of the crucial physiologic conditions leading to UPR activation, which is associated with several human diseases including tissue ischemia and cancer [5]. H9c2 cells were subjected to a combination of serum and glucose deprivation as described in materials and methods. We observed that glucose deprivation induced the expression of UPR target genes GRP78 and HERP, thereby confirming induction of the UPR (Figure 1B). We found that the conditions of glucose deprivation decreased the levels of primary miR-17-92 transcript in H9c2 cells (Figure 1B). In addition, treatment of MDA-MB231 cells (Figure 1C) and HEK-293T cells (Figure 1D) with TG or TM led to the induction of UPR-target genes (GRP78 and HERP) and downregulation of the primary miR-17-92 transcript. Furthermore, TM treatment of H9c2 cells showed a decrease in the expression of primary miR-17-92 transcript as well as all the six miRNAs belonging to the miR-17-92 cluster (Figure 2A,B). In this study, we have observed reduced expression of the primary miR-17-92 transcript utilising a variety of cell lines and methods to induce the UPR. As such our results suggest that downregulation of the miR-17-92 cluster in response to ER stress is not cell-type or stimulus dependent.

### 3.2. Repression of miR-17-92 Promoter during ER Stress is PERK-Dependent

Expression of the miR-17-92 cluster has been shown to be regulated at the transcriptional level in several experimental models [24,26]. To elucidate the mechanism of downregulation of the miR-17-92 cluster during conditions of ER stress we performed miR-17-92 promoter reporter assays. For this purpose, we used a reporter construct accommodating a ~3700 bp DNA fragment containing the genomic locus of the miR-17-92 cluster cloned at the 5′ site of the luciferase gene in the reporter pGL4.10 vector (pGL4-17-92 FL) (Figure 3A) [29]. As shown in Figure 3B, pGL4-17-92 FL reporter activity was reduced in MCF-7 cells treated with TG or TM, but no reduction in pGL4-17-92 FL reporter activity was observed upon treatment with hydrogen peroxide. The control plasmid (pGL4.10) gave much lower activity and did not show reduction in reporter activity upon treatment with TG or TM (data not shown). To explore the mechanism behind miR-17-92 cluster downregulation during ER stress we used PERK-K618A, a dominant-negative PERK mutant that blocks transphosphorylation and activation of the PERK kinase [36,37]. Co-expression of PERK-K618A completely inhibited the ER stress-mediated downregulation of miR-17-92 promoter activity (Figure 3C). The wild-type and K618A PERK proteins were expressed at comparable levels. These data suggest that the PERK arm of the UPR is required for repression of the miR-17-92 cluster during conditions of ER stress. The transcription factors ATF4, NRF2 and CHOP are activated following PERK activation during ER stress [1,23]. Next, we determined the effect of ATF4, NRF2 and CHOP co-expression on the pGL4-17-92 FL reporter activity. Co-transfection of the pGL4-17-92 FL construct with an expression vector for ATF4 or NRF2 in MCF-7 cells led to a significant reduction in luciferase activity, whereas co-expression of CHOP had no such effect (Figure 3D). The expressions of ectopic ATF4, NRF2 and CHOP were comparable during these experiments. Co-expression of ATF4, NRF2 and CHOP had no effect on the activity of control empty vector (pGL4.10) (data not shown).

To map the region in the miR-17-92 promoter that responds to the ER stress-mediated repression, we used promoter constructs with different lengths of the miR-17-92 5′ flanking region (−829 bp to +2922 bp (17-92FL); +1492 bp to +2922 bp (17-92/1); +2626 bp to +2922 bp (17-92)) cloned into the promoterless luciferase pGL4 vector (Figure 4A) [26,29,30]. When transfected cells were exposed to ER stress conditions, the activities of the 17-92FL, 17-92/1 and 17-92 promoter constructs were greatly reduced in MCF-7 cells (Figure 4B). Next, we determined the effect of ATF4 and NRF2 co-expression on the activity of 17-92FL, 17-92/1 and 17-92 reporter constructs. We observed significant reduction in the activity of 17-92FL, 17-92/1 and 17-92 promoter constructs upon co-expression of ATF4 or NRF2 (Figure 4C). These results suggest the presence of a cis-regulatory element responsive to ATF4 and NRF2 in the promoter region of the miR-17-92 cluster at position +2626 to +2922 (17-92) relative to the transcription start site.

We next used the photosensitizer hypericin, which accumulates preferentially in the ER membrane and upon light exposure generates reactive oxygen species (ROS), causing a loss-of function of the Ca^2+^-ATPase pump (SERCA) and consequent ER stress [38]. Hypericin-mediated photo-oxidative (phOx) ER damage induces PERK-dependent cell death [39]. We investigated the effect of phOx-mediated ER stress on the expression level of representative miRNAs belonging to the miR-17-92 cluster in wild-type and PERK^−/−^ MEFs. We found that phOx-mediated ER stress led to a significant decrease in the levels of miR-17, miR-18a and miR-20a in wild-type but not in PERK^−/−^ MEFs (Figure 5A). Next, we determined the effect of ectopic ATF4 and NRF2 on the expression level of representative miRNAs belonging to the miR-17-92 cluster (miR-17, miR-18a and miR-19a). We found that overexpression of both ATF4 and NRF2 in PC12 cells led to a significant decrease in the levels of miR-17, miR-18a and miR-20a (Figure 5B,C). Taken together, these results show that ATF4 and NRF2 expression can repress the expression of miRNAs belonging to the miR-17-92 cluster independent of ER stress. To elucidate the molecular mechanism of miR-17-92 repression, we transfected MCF-7 cells with GFP, ATF4 or NRF-2 expression plasmids and performed ChIP assays to evaluate their interaction with promoters of miR-17-92, heme oxygenase-1 (HO-1) and CHOP. The ChIP assay confirmed the recruitment of NRF2 to the promoter of HO-1, a bonafide NRF2-responsive gene (Appendix A, in the Appendix A) [40]. Further, ATF4 was recruited to the CHOP promoter, a known ATF4-responsive gene (Appendix A). However, under similar conditions we did not observe recruitment of ATF4 and/or NRF2 to the promoter region of miR-17-92 (Appendix A). Our results (RT-PCR and promoter reporter assays) suggest that ATF4 and NRF2 downregulate the expression of miR-17-92 cluster, but the molecular mechanism for the repression of miR-17-92 by ATF4 and NRF2 is not clear and needs further investigation.

### 3.3. MiR-17-92 Cluster Protects Against UPR-Induced Cell Death

To determine the role of the miR-17-92 cluster on ER stress-induced apoptosis, we evaluated the effect of miR-17-92 cluster overexpression on ER stress-induced apoptosis. For this purpose, SH-SY5Y and H9c2 cells were transduced with lentivirus engineered to produce RFP and miR-17-92. The control and miR-17-92 overexpressing clones of SH-SY5Y (Figure 6A) and H9c2 (Figure 6C) showed expression of RFP. The miR-17-92 expressing clones of SH-SY5Y (Figure 6B) and H9c2 (Figure 6D) cells showed increased expression of miRNAs belonging to the miR-17-92 cluster as compared to the control clones. However, we observed variations in the level of individual miRNAs in the miR-17-92 expressing clones of SH-SY5Y (Figure 6B) and H9c2 (Figure 6D) cells suggesting cell-type dependent processing of the primary transcript. Next, we evaluated whether expression of the miR-17-92 cluster can affect sensitivity to ER stress-induced cell death. We found that ER stress-induced apoptosis was attenuated in miR-17-92 expressing clones of SH-SY5Y (Figure 7A) and H9c2 (Figure 7C) cells as compared with control clones. Western blot analysis revealed that treatment with TG and TM induced processing of caspase-3 in control and miR-17-92 overexpressing clones of SH-SY5Y (Figure 7B) and H9c2 (Figure 7D) cells. Notably there was decreased processing of caspase-3 in the miR-17-92 expressing cells as compared to control cells (Figure 7B,D). Thus, overexpression of miR-17-92 cluster provided resistance against ER stress-induced apoptosis.

## 4. Discussion

There is growing evidence showing a close functional interaction between ER stress response and miRNAs. Here we have shown that expression of miRNAs belonging to the miR-17-92 cluster is markedly downregulated during ER stress conditions. The repression of miR-17-92 cluster during ER stress is dependent on PERK signalling. Several studies have reported a cross-talk between UPR and miRNAs in regulation of the UPR signalling pathway, and determining cell fate during conditions of UPR [19,41]. The RIDD activity of IRE1 results in the degradation of a sub-set of miRNAs that attenuate caspase-2 mRNA translation. This leads to an increase and activation of caspase-2 protein, an initiator protease of the cell death pathway [42]. In addition, the RIDD activity of IRE1 can reduce the expression of many tumour suppressor miRNAs, such as miR-34 in acute myeloid leukaemia and miR-3609 in breast cancer, to promote cancer progression [43,44]. We have previously shown that PERK-mediated downregulation of the miR-424(322)-503 cluster regulates optimal activation of IRE1 and ATF6 during conditions of ER stress [45]. Downstream of PERK signalling, the ATF4-mediated induction of miR-211 and NRF2-mediated repression of miR-214 has been shown to promote cell survival [46]. We have shown that ER stress-induced expression of miR-7a can promote cell survival via indirect repression of CHOP, a key pro-apoptotic molecule in the UPR [16]. We have additionally shown that during conditions of ER stress both ATF4 and NRF2 downregulate expression of the miR-106b-25 cluster and increase UPR-mediated cell death by increasing the expression of Bcl-2 homology domain 3 (BH3)-only protein, BIM [47]. Here our results show that ectopic expression of ATF4 or NRF2 can lead to a decrease in the levels of miRNAs comprising the miR-17-92 cluster as well as a reduction in miR-17-92 promoter reporter activity. Furthermore, the miR-17-92 and miR-106b-25 clusters have been shown to be transcriptionally regulated by the MYC and E2F family of transcription factors [24,30]. The expressions of both the miR-17-92 and miR-106b-25 clusters are downregulated during the UPR in a PERK-dependent manner.

Transcriptional regulation of gene expression (activation and repression) is essential for cell growth, differentiation and death in metazoans. Compared to gene activation, the molecular mechanisms that lead to gene repression are not well studied. Understanding transcriptional repression is equally important because defects in gene repression may result in developmental abnormalities and diseases [48]. Our results (RT-PCR and promoter reporter assays) show that ATF4 and NRF2 downregulate expression of the miR-17-92 cluster but that the molecular mechanisms for the repression of miR-17-92 by ATF4 and NRF2 are not clear and requires further study. ATF4 contains a basic-leucine-zipper domain and forms heterodimers with several interacting partners such as the activator protein 1 (AP-1) and CCAAT-enhancer binding protein (C/EBP) family [49]. In concert with interacting partners, ATF4 upregulates expression of specific genes via a C/EBP-activating transcription factor (C/EBP-ATF) response element [49]. ATF4 has been described as both a negative [50] and a positive regulator [51] of cAMP response elements (CRE)-dependent transcription. ATF4 has been shown to negatively regulate the CRE of the Melanocyte Inducing Transcription Factor (MITF) promoter upon glucose deprivation in melanoma [52]. Further, ATF4 forms a heterodimer with Disrupted-in-Schizophrenia 1 (DISC1) after which the ATF4-DISC1 heterodimer is recruited to the promoter of Phosphodiesterase 4D (PDE4D) where it represses transcription [53]. NRF2 has been reported to induce the expression of several hundred target genes by dimerizing with small musculoaponeurotic fibrosarcoma (sMAF) transcription factors (sMAFs) and triggering the recruitment of coactivator complexes [49,54]. However, NRF2 interacts with replication protein A1 (RPA1) and the NRF2-RPA1 heterodimer can transcriptionally repress gene expression [55]. For example, downregulation of myosin light chain kinase (MYLK) is mediated by the recruitment of the NRF2-RPA1 complex at its promoter [55]. In this context, RPA1 competes with sMAF for NRF2 binding and NRF2-sMAF-containing transcriptional activator complex is replaced by an NRF2-RPA1-containing repressor complex that results in repression of MYLK [55]. In conclusion, both ATF4 and NRF2 can positively and negatively regulate gene expression and additional studies are required to determine the molecular details of miR-17-92 promoter repression by ATF4 and NRF2.

The three UPR sensors can induce or suppress miRNAs, which has implications for cell fate during conditions of ER stress [19]. The dysregulation of miRNAs is likely to have pleiotropic effects and to contribute to the pathogenesis of diseases, including cancer [24,56]. The functional role of miR-17-92 during ER stress has been demonstrated by overexpression experiments, whereby increased expression of the miR-17-92 cluster inhibited ER stress-induced apoptosis in neuroblastoma cells and cardiomyocytes. Indeed, in neuroblastomas, miR-17-92 has shown oncogenic activity by repressing the expression of key molecules within the Transforming growth factor beta (TGF-β) signalling pathway [57]. Further, overexpression of miR-17-92 in adult cardiomyocytes protects the heart from myocardial infarction-induced injury [58]. Although several studies have characterised the oncogenic activities of miR-17-92, loss of heterozygosity at the miR-17-92 locus has been reported [22]. The ectopic expression of miR-17-92 exhibits tumour suppressive activity and promotes drug-sensitivity in prostate cancer cells [59]. The transgenic miR-17-92 expression in epithelial cells of the small and large intestine inhibits cancer progression by impairing tumour angiogenesis [60]. The miR-17-92 cluster exhibits a context-dependent role in subtypes of breast cancer where it provides a tumour suppressive function in ER-positive breast cancer and shows oncogenic effects in triple negative breast cancer [23]. Furthermore, ectopic expression of the miR-17-92 cluster sensitised MCF-7 cells to chemotherapeutic compounds, whereas miR-17-92 expressing SKBR3 cells showed resistance to them [23]. Taken together, these observations suggest that miR-17-92 has both oncogenic and tumour suppressor functions under different conditions. As a result that miRNAs can target many distinct mRNA targets, the physiological consequence of expression is likely to be context and/or tissue-dependent. Therefore, the functional consequences of UPR-mediated miR-17-92 cluster downregulation in human cancers is likely to be context dependent.

What is the biological significance of transcriptional repression of the miR-17-92 cluster by ATF4 and NRF2? The transcription factors ATF4 and NRF2 are activated during ER stress in a PERK-dependent manner [46]. ATF4 controls the expression of a wide range of adaptive genes that allow cells to endure periods of stress, such as hypoxia or amino acid limitation [49]. However, under persistent stress conditions, ATF4 promotes the induction of apoptosis [61]. ATF4 has been shown to act as a pro-death transcriptional regulator in the nervous system that propagates death responses to oxidative stress in vitro and to stroke in vivo [62]. ATF4 also mediates the ER stress-induced cell death of neuroectodermal tumour cells in response to fenretinide or bortezomib [63]. NRF2 is a key transcription factor that controls the cascade of cytoprotective and antioxidant defence mechanisms, and the maintenance of redox homeostasis [6,54]. Further, sustained activation of NRF2 in ATG5-deficient mouse livers due to p62 mediated stabilisation of NRF2 has been reported to be a major cause of toxicity in autophagy-impaired livers [64]. In addition, sulforaphane-induced NRF2 was reported to trigger a caspase3/7-independent and p38 MAPK-dependent cell death in mycobacteria infected macrophages [65]. However, the molecular mechanisms by which ATF4 and NRF2 exert their pro-apoptotic effects during the UPR are not yet clearly understood. The data presented here provides a new molecular mechanism underlying the ATF4 and NRF2-mediated induction of cell death during conditions of ER stress.

In summary, our study shows PERK-dependent downregulation of the miR-17-92 cluster during conditions of ER stress. We found that levels of the miR-17-92 primary transcript, as well mature miRNAs belonging to this cluster, were reduced upon induction of ER stress. Furthermore, our results revealed that transcriptional repression of the miR-17-92 cluster during the UPR is mediated by ATF4 and NRF2 downstream of PERK signalling. However, further work is required to elucidate the molecular mechanisms underlying repression of the miR-17-92 promoter by ATF4 and NRF2. Finally, we have shown that ectopic expression of miR-17-92 provided resistance towards UPR-induced cell death and suggest that miR-17-92 repression during the UPR can regulate cell fate decisions during conditions of ER stress.

## Figures and Tables

**Figure 1 life-11-00030-f001:**
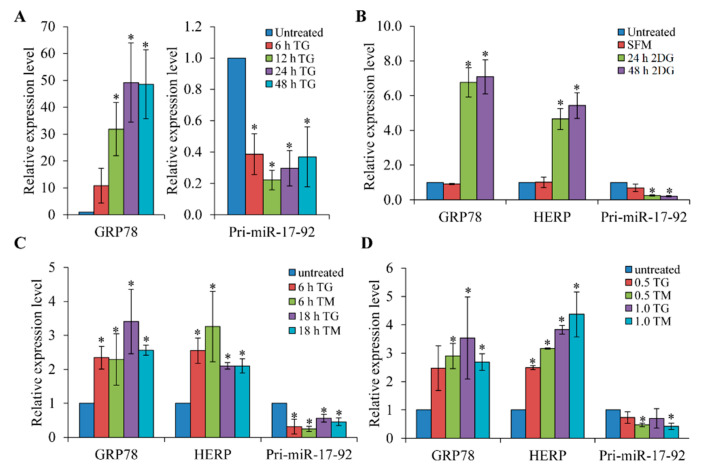
Downregulation of primary miR-17-92 transcript during conditions of ER stress. (**A**) H9c2 cells were either untreated or treated with (1.0 μM) thapsigargin (TG) and the expression levels of GRP78 and primary miR-17-92 transcripts were quantified by qRT-PCR, normalising against GAPDH (*n* = 3). (**B**) H9c2 cells were treated with 2-deoxyglucose (1 mM) along with serum deprivation for 24 and 48 h. The expression level of GRP78, HERP and primary miR-17-92 transcript was quantified by qRT-PCR, normalising against GAPDH (*n* = 3). Serum free medium (SFM); 2-deoxyglucose (2DG). (**C**) MDA-MB231 cells were either untreated or treated with (1.0 μM) thapsigargin (TG) and (0.5 μg/mL) tunicamycin (TM) and expression level of GRP78, HERP and primary miR-17-92 transcript was quantified by qRT-PCR, normalising against GAPDH (*n* = 3). (**D**) HEK-293T cells were either untreated or treated for 24 h with (0.5 and 1.0 μM) thapsigargin (TG) and (0.5 μg/mL and 1.0 μg/mL) tunicamycin (TM) and expression level of GRP78, HERP and primary miR-17-92 transcripts was quantified by qRT-PCR, normalising against GAPDH (*n* = 3). * *p* < 0.05, two-tailed paired *t*-test compared with untreated cells.

**Figure 2 life-11-00030-f002:**
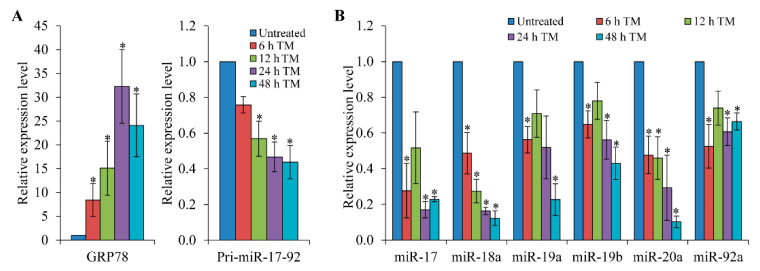
Downregulation of miRNAs belonging to miR-17-92 cluster during conditions of ER stress. Total RNA was isolated from H9c2 cells that were either untreated or treated with (1.0 μg/mL) tunicamycin (TM) for the indicated time points. (**A**) The expression level of GRP78 and primary miR-17-92 transcript was quantified by qRT-PCR, normalising against GAPDH (*n* = 3). (**B**) The expression levels of member miRNAs of the miR-17-92 cluster were quantified by real-time RT-PCR, normalising against snoRNA (*n* = 3). * *p* < 0.05, two-tailed paired *t*-test compared with untreated cells.

**Figure 3 life-11-00030-f003:**
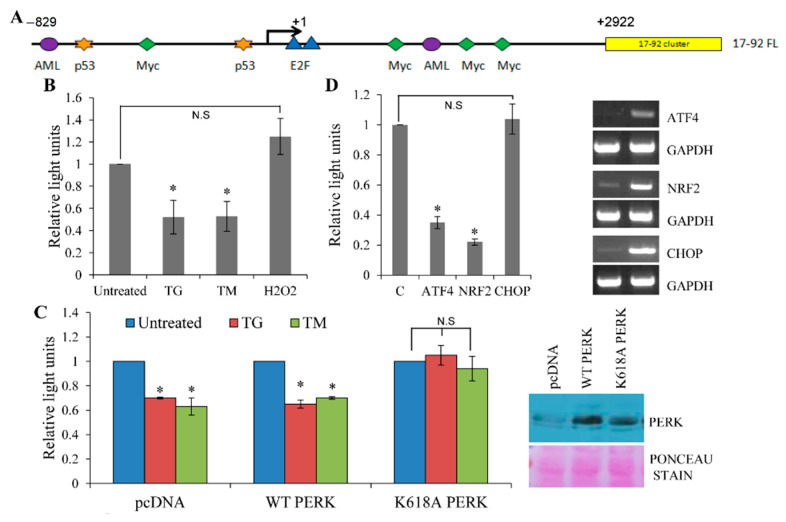
Repression of miR-17-92 promoter during ER stress is PERK dependent. (**A**) Schematic representation of the genomic region of the miR-17-92 cluster. Arrow indicates the transcription start site. (**B**) After transfection with pGL4-17-92 FL reporter MCF-7 cells were treated with (2.0 μM) thapsigargin (TG) and (2.0 μg/mL) tunicamycin (TM) or (600 μM) hydrogen peroxide (H_2_O_2_). Normalised luciferase activity (Firefly/*Renilla*) is shown (*n* = 3). (**C**) MCF-7 cells were transfected with pGL4-17-92 FL in combination with control (pcDNA), wild-type PERK (WT PERK) or dominant-negative PERK (K618A PERK) expression plasmids. 24 h post-transfection cells were left untreated or treated with (2.0 μM) thapsigargin (TG) and (2.0 μg/mL) tunicamycin (TM). Normalised luciferase activity (Firefly/*Renilla*) is shown (*n* = 3). Cell lysates were analysed by western blotting using PERK antibody. Ponceau staining is shown as a loading control. (**D**) MCF-7 cells were transfected with pGL4-17-92 FL in combination with the control pcDNA3 (**C**), wild-type ATF4, NRF2 or CHOP expression plasmids. Normalised luciferase activity (Firefly/*Renilla*) is shown (*n* = 3). * *p* < 0.05, two-tailed paired *t*-test compared with control cells. Expression of ectopic ATF4, NRF2 and CHOP was evaluated by RT-PCR.

**Figure 4 life-11-00030-f004:**
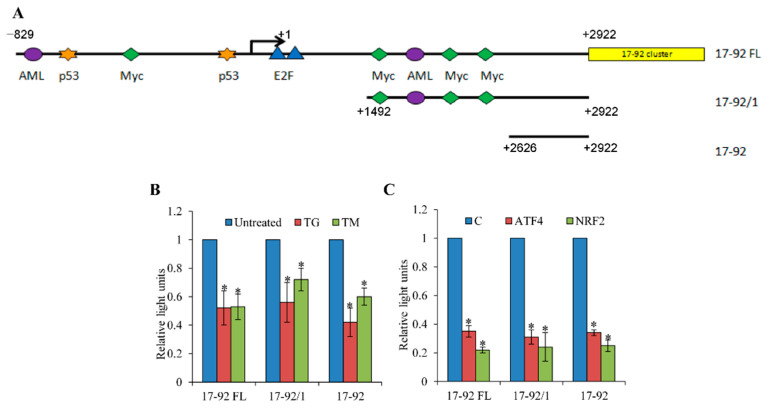
Mapping the region within the miR-17-92 promoter that responds to ATF4 and NRF2 -mediated repression. (**A**) Schematic representations of the miR17-92 cluster promoter deletion constructs are shown. (**B**) MCF-7 cells were transfected with the indicated miR-17-92 promoter reporter constructs and 24 h post-transfection cells were either left untreated or were treated with (2.0 μM) thapsigargin (TG) and (2.0 μg/mL) tunicamycin (TM). Normalised luciferase activity (Firefly/*Renilla*) is shown (*n* = 3). (**C**) MCF-7 cells were transfected with the indicated miR-17-92 reporter construct in combination with the control pcDNA3 (**C**), wild-type ATF4 or NRF2 expression plasmids. Normalised luciferase activity (Firefly/*Renilla*) is shown (*n* = 3).

**Figure 5 life-11-00030-f005:**
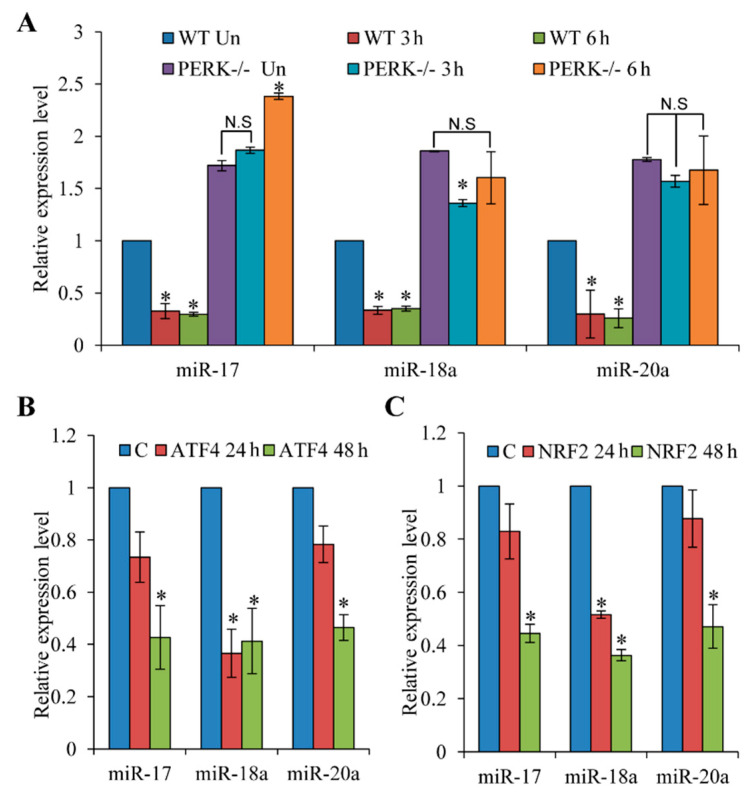
PERK-dependent regulation of the miR-17-92 cluster during conditions of ER stress. (**A**) Wild-type (WT) and PERK knockout (PERK^−/−^) MEFs were exposed to phOx stress (200 nM hypericin for 2 h, 2.7 J/cm^2^). Expression levels of miR-17, miR-18a and miR-20a were quantified by real-time RT-PCR, normalising against snoRNA (*n* = 2). (**B**) PC12 cells were transfected with the control pcDNA3 **(C)**, wild-type ATF4 or NRF2 expression plasmids and total RNA was isolated at the indicated time points. Expression levels of miR-17, miR-18a and miR-20a were quantified by real-time RT-PCR, normalising against snoRNA (*n* = 3). * *p* < 0.05, Two-tailed paired *t*-test compared to control samples.

**Figure 6 life-11-00030-f006:**
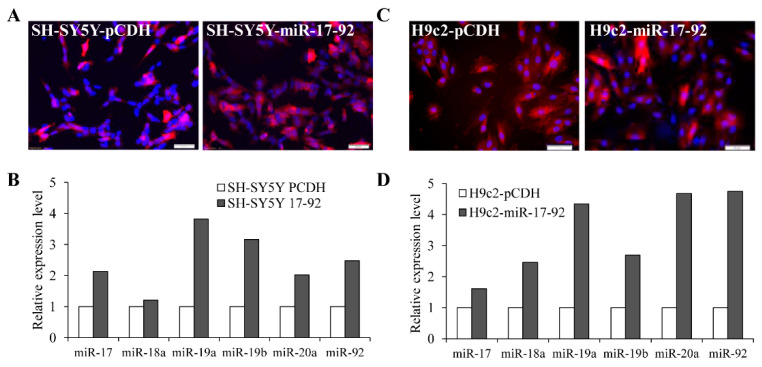
Generation and characterisation of the stable clones expressing the miR-17-92 cluster. (**A**) Photomicrographs of SH-SY5Y-PCDH and SH-SY5Y-miR-17-92 cells stained with DAPI. (**B**) Expression level of miRNAs belonging to miR-17-92 cluster in SY5Y-PCDH and SH-SY5Y-miR-17-92 cells were quantified by qRT-PCR, normalising against snoRNA. (**C**) Photomicrographs of H9c2-PCDH and H9c2-miR-17-92 cells stained with DAPI. (**D**) Expression levels of miRNAs belonging to the miR-17-92 cluster in H9c2-PCDH and H9c2-miR-17-92 cells and were quantified by qRT-PCR, normalising against snoRNA. Scale bar is 50 μm.

**Figure 7 life-11-00030-f007:**
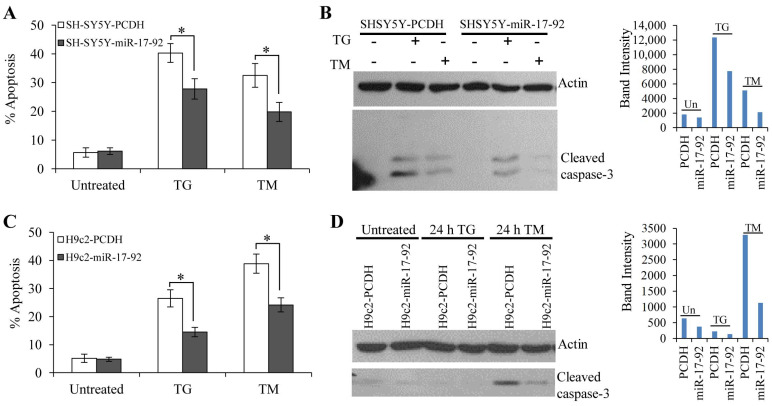
Effect of the miR-17-92 cluster on UPR-mediated cell death. (**A**,**B**) SH-SY5Y-PCDH and SH-SY5Y-miR-17-92 cells were treated with (0.5 μM) TG and (0.5 μg/mL) TM for 24 h. (**A**) After treatment the apoptotic nuclei were determined as described in Materials and Methods (*n* = 3). (**B**) Immunoblots for cleaved caspase-3 and actin are shown. Expression level of cleaved caspase-3 relative to actin was determined using ImageJ. (**C**,**D**) H9c2-PCDH and H9c2-miR-17-92 cells were treated with (1.0 μM) TG and (1.0 μg/mL) TM for 24 h. (**C**) After treatment the apoptotic nuclei were determined as described in Materials and Methods (*n* = 3). (**D**) Immunoblots for cleaved caspase-3 and actin are shown. * *p* < 0.05, two-tailed paired *t*-test compared with control cells.

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
