# Peer review of "Downregulation of miR-17-92 Cluster by PERK Fine-Tunes Unfolded Protein Response Mediated Apoptosis"

_life, 2021, doi:10.3390/life11010030_

Round 1

Reviewer 1 Report

The author investigated the relation between UPR and expression of miR-17-92 clusters and proved that downregulation of these miRs is PERK dependent.

Results and Figures:

The experimental detail should be listed in the Material and Methods, not in the Figure Legend (Fig1, 3, 4, 5, 6). Figure legend only describe the figure data. Do not need to repeat "experiment data represent mean+_SD from three independent experiments." under each figure, as long as it shows in the "statistical analysis" of Material and Methods.

Plasmid constructs:

Is the Prom17M the name of vector? Or "promoter of miR-17-92"? At least, the author need to provide the complete construct name like "pCXN2-miR17-92" as in the lentivirus part.

Line113, the subclones expressing miR-106b-25?

RNA extraction and qpcr:

Have you considered the miRNeasy kit from Qiagen? This may provide higher recovery for small RNAs, but regular RNeasy should also work.

Provide catalog# for miR primer or assay.

Result:

3.1 List microarray data in the supplymental or provide access number.

Figure1D, missing ug/mL in figure legend.

3.2 Need to explain why luciferase activity do not reduce after cotransfection of CHOP.

Figure4D, what's the -ve control? Why there are multiple bands in CHOP lane and even in ve control?

Figure5A, it looks like there is siganificant difference between PERK-/-un and PERK-/-3h in the miR-18a group based on the mean and SD bar shows here.

Figure6H, There is cleaved caspase3 band even in the untreated group but not in the 24h TG. And the western blot result need to quantified by the intensity of the band.

Typo and style:

Line 41, missing full name of ERAD.

Some superscript and subscript need to be corrected, as line 145, 167, 187.

Line 188, 193, degree.

Author Response

Comments and Suggestions for Authors:

The author investigated the relation between UPR and expression of miR-17-92 clusters and proved that downregulation of these miRs is PERK dependent.

Author’s response: We thank the reviewer for the appreciation and encouraging comments.

Results and Figures:

The experimental detail should be listed in the Material and Methods, not in the Figure Legend (Fig1, 3, 4, 5, 6). Figure legend only describe the figure data. Do not need to repeat "experiment data represent mean+_SD from three independent experiments." under each figure, as long as it shows in the "statistical analysis" of Material and Methods.

Author’s response: We thank the reviewer for this comment. We have made suggested changes to the figure legends of figure 1, 3, 4, 5, & 6.  Specifically we have removed experimental details from the figure legends.

Plasmid constructs:

Is the Prom17M the name of vector? Or "promoter of miR-17-92"? At least, the author need to provide the complete construct name like "pCXN2-miR17-92" as in the lentivirus part.

Author’s response: We thank the reviewer for this feedback. The vector name (pGL4) all the miR-17-92 promoter reporter constructs has been added in the methods section (L126- 129 of revised manuscript).

Line113, the subclones expressing miR-106b-25?

Author’s response: We thank the reviewer for pointing out this mistake. This was a typographical error and has been corrected to subclones expressing miR-17-92.

RNA extraction and qpcr:

Have you considered the miRNeasy kit from Qiagen? This may provide higher recovery for small RNAs, but regular RNeasy should also work.

Author’s response: We have compared the recovery of small RNAs using RNAeasy kit as well as Trizol with a modification where 0.75 ml isopropanol is added to ml TRIzol, mix and kept O/N at -20 degrees for precipitation. We found that yield of small RNA were comparable between RNAeasy and modified Trizol method.

Provide catalog# for miR primer or assay.

Author’s response: We thank the reviewer for this feedback. The catalog# for all the miRNAs have been added in the materials and methods section of revised version (L163 to 166 of revised manuscript)

Result:

3.1 List microarray data in the supplemental or provide access number.

Author’s response: We thank the reviewer for this feedback. The miRNA microarray data is provided as supplementary table 1 of the revised manuscript.

Figure1D, missing ug/mL in figure legend.

Author’s response: We thank the reviewer for this feedback. The suggested change has been made in the figure legend of revised manuscript.

3.2 Need to explain why luciferase activity do not reduce after cotransfection of CHOP.

Figure4D, what's the -ve control? Why there are multiple bands in CHOP lane and even in ve control?

Author’s response: We thank the reviewer for this comment. This is likely due to absence of CHOP binding sites in the reporter constructs and/or lack of functional interaction between CHOP and miR-17-92 promoter. The -ve control is no template control for PCR reaction. The bands in -ve control are likely due to formation of primer dimer. A Primer dimer (PD) is a by-product in PCR, it consists of primer molecules that have attached (hybridized) to each other because of strings of complementary bases in the primers.

Figure5A, it looks like there is significant difference between PERK-/-un and PERK-/-3h in the miR-18a group based on the mean and SD bar shows here.

Author’s response: We thank the reviewer for this feedback. We apologize for this error in the original manuscript and this has been corrected in the revised manuscript.

Figure6H, There is cleaved caspase3 band even in the untreated group but not in the 24h TG. And the western blot result need to quantified by the intensity of the band.

Author’s response: We thank the reviewer for this comment. We observed varying amount of cleaved caspase-3 in the untreated group during the different repeats and this is reflection of basal apoptosis in response to homeostatic and environmental stress during the culturing conditions. Notably miR17-92 expressing sub-clone show reduced basal and UPR-induced levels of cleaved caspase-3 levels. As suggested by reviewer we quantified the intensities of cleaved caspase-3 using imageJ and added to the Fig 7B & D of revised manuscript.

Typo and style:

Line 41, missing full name of ERAD.

Some superscript and subscript need to be corrected, as line 145, 167, 187.

Line 188, 193, degree.

Author’s response: We thank the reviewer for this feedback. The suggested change has been made in the revised manuscript.

Reviewer 2 Report

This is an interesting manuscript reporting the involvement of PERK-dependent downregulation of miR-17-92 in the promotion of ER stress-induced apoptosis. The following concerns should be addressed:

1) Introduction: NRF2 and the effects of its activation should be mentioned, including its direct activation through its phosphorylation by PERK.

2) Results, starting L. 202: The “preliminary experiments” should be shown, at least in a supplementary figure. 

3) Fig. 3: Should show expression immunoblots of PERK wt and mutant, ATF4, NRF2, CHOP

4) L. 266 and Fig. 4D: The image shows a very weak and not convincing result of NRF2 binding. This should be repeated and shown more convincingly and it should be indicated how many times this experiment was done. Gels with the different miR17-92 cluster promoter deletion constructs should be shown.

5) Fig. 6 D and H should be quantified.

6) Discussion, starting L. 330. The discussion mentions only the pro-apoptotic effects of ATF and NRF2. However, they also have cell protective effects, and there is an extensive literature on this. The cell protective effects should be mentioned and discussed in the context of the article.  Why do their pro-apoptotic effects seem to be prevalent in this study? Perhaps there are also effects on some other factor, which counteracts the effects on miR-17-92? 

7) Fig. 6. Fonts are too small

Author Response

Comments and Suggestions for Authors

This is an interesting manuscript reporting the involvement of PERK-dependent downregulation of miR-17-92 in the promotion of ER stress-induced apoptosis.

Author’s response: We thank the reviewer for the appreciation and encouraging comments.

The following concerns should be addressed:

1) Introduction: NRF2 and the effects of its activation should be mentioned, including its direct activation through its phosphorylation by PERK.

Author’s response: We thank the reviewer for this feedback. We have added the following sentences in the introduction of the revised manuscript (L57 to L61). “Nuclear factor (erythroid-derived 2)-like 2 (NRF2), a b-ZIP cap “n” collar transcription factor can also be induced by PERK. NRF2 is normally bound to its negative regulator Kelch-like ECH-associated protein (KEAP1) in the cytoplasm (Kobayashi and Yamamoto, 2005; Lee et al., 2007). In addition activated PERK phosphorylates NRF2 leading to its dissociation from KEAP1 (Cullinan et al., 2003) and translocation to the nucleus where it binds to antioxidant response element (ARE) in the promotors of its target genes.”

2) Results, starting L. 202: The “preliminary experiments” should be shown, at least in a supplementary figure. 

Author’s response: We thank the reviewer for this feedback. The miRNA microarray data is provided as Supplementary Table 1 of the revised manuscript.

3) Fig. 3: Should show expression immunoblots of PERK wt and mutant, ATF4, NRF2, CHOP

Author’s response: We have added data showing the expression WT and K618A mutant PERK in Fug 3C of the revised manuscript. We have shown previously that expression plasmids for ATF4 and NRF2 express the desired protein [Figure 2 of the paper- Oncogene. 2016 Nov 10; 35(45): 5860–5871]. For reporter assay experiments we have evaluated the expression of ectopic ATF4, NRF2 and CHOP by conventional RT-PCR. This RT-PCR data has been added to Fig 3D ofthe revised manuscript.

4) L. 266 and Fig. 4D: The image shows a very weak and not convincing result of NRF2 binding. This should be repeated and shown more convincingly and it should be indicated how many times this experiment was done. Gels with the different miR17-92 cluster promoter deletion constructs should be shown.

Author’s response: We thank the reviewer for this concern. We agree with the reviewer the ChIP-PCR do not convincingly show binding of ATF4 and/or NRF2 to the promoter region of miR-17-92. Accordingly we have made changes to the revised manuscript (L285 to L293). “The ChIP assay confirmed the recruitment of NRF2 to the promoter of HO-1, a bonafide NRF2-responsive gene (Supplementary Figure 1) [40]. ATF4 was recruited to the CHOP promoter (Supplementary Figure 1). CHOP, a known UPR-responsive gene, is confirmed here to be regulated by ATF4. However, under similar conditions we did not observe recruitment of ATF4 and/or NRF2 to the promoter region of miR-17-92 (Supplementary Figure 1). Our results (RT-PCR and promoter reporter assays) suggest that ATF4 and NRF2 downregulate the expression of miR-17-92 cluster but the molecular mechanism for the repression of miR-17-92 by ATF4 and NRF2 is not clear and needs further investigation..”  

5) Fig. 6 D and H should be quantified.

Author’s response: We thank the reviewer for this feedback. As suggested by reviewer we quantified the intensities of cleaved caspase-3 using imageJ and added to Fig 7 B &D of the revised manuscript.

6) Discussion, starting L. 330. The discussion mentions only the pro-apoptotic effects of ATF and NRF2. However, they also have cell protective effects, and there is an extensive literature on this. The cell protective effects should be mentioned and discussed in the context of the article.  Why do their pro-apoptotic effects seem to be prevalent in this study? Perhaps there are also effects on some other factor, which counteracts the effects on miR-17-92? 

Author’s response: We thank the reviewer for this feedback. We have rewritten the discussion in the revised manuscript. Specifically description on the transcriptional activation and suppression functions of ATF4 and NRF2 (L343 to L359) and pro-survival and pro-death effects of ATF4 and NRF2 (L386 to L399) are added to the discussion

7) Fig. 6. Fonts are too small

Author’s response: We thank the reviewer for this feedback. We have increased the font size for figure legend of Fig 6.

Reviewer 3 Report

Authors demonstrated the mechanism of PERK signalling related cell death via NRF2 mediated repression of miR17-92 cluster. Some parts should be revised.

  1. Scale bar should be added in figure 6A and E.
  2. The resolution of overall figures should be enhanced.
  3. Discussion section should contain detailed vision of the authors regarding these results.
  4. All of the abbreviations should be explain at their first appearances.
  5. The providers of products purchased should be explained as (company name, state name, country name) in materials and methods section.

Author Response

Authors demonstrated the mechanism of PERK signalling related cell death via NRF2 mediated repression of miR17-92 cluster.

Author’s response: We thank the reviewer for the appreciation and encouraging comments.

Some parts should be revised.

  1. Scale bar should be added in figure 6A and E.

Author’s response: We thank the reviewer for this feedback. The scale bar has been added to fig 6 A & B of the revised manuscript.

  1. The resolution of overall figures should be enhanced.

Author’s response: We thank the reviewer for this feedback. The suggested change has been made in the revised manuscript.

  1. Discussion section should contain detailed vision of the authors regarding these results.

Author’s response: We thank the reviewer for this feedback. We have rewritten the discussion in the revised manuscript.

  1. All of the abbreviations should be explain at their first appearances.

     Author’s response: The suggested change has been made to the revised manuscript.

  1. The providers of products purchased should be explained as (company name, state name, country name) in materials and methods section.

Author’s response: The suggested change has been made to the revised manuscript.

Round 2

Reviewer 2 Report

All my concerns have been addressed. The manuscript is much improved 

Author Response

We thank the reviewer for the appreciation and encouraging comments.